# Dual-responsive Gemini Micelles for Efficient Delivery of Anticancer Therapeutics

**DOI:** 10.3390/polym11040604

**Published:** 2019-04-02

**Authors:** Young In Choi, Eun-sook Choi, Kwan Ho Mun, Se Guen Lee, Sung Jun Lee, Sang Won Jeong, Seung Woo Lee, Hyun-Chul Kim

**Affiliations:** 1Convergence Research Institute, Daegu Gyeongbuk Institute of Science and Technology (DGIST), Daegu 42988, Korea; cyi9161@dgist.ac.kr (Y.I.C.); stom96@dgist.ac.kr (E.-s.C.); sklee@dgist.ac.kr (S.G.L.); schrisj@dgist.ac.kr (S.J.L.); sjeomg@dgist.ac.kr (S.W.J.); 2School of Chemical Engineering, Yeungnam University, Gyeongsan 38541, Korea; kwanho0903@gmail.com

**Keywords:** gemini micelles, drug delivery systems, disulfide–thiol exchange, reactive oxygen species

## Abstract

Polymeric micelles as drug delivery vehicles are popular owing to several advantages. In this study, a gemini amphiphile (gemini mPEG-Cys-PMT) consisting of hydrophilic poly(ethylene glycol) and hydrophobic poly(methionine) with cystine disulfide spacer was synthesized and its micellar properties for thiol- or reactive oxygen species (ROS)-dependent intracellular drug delivery were described. The cleavage of cystine linkage in a redox environment or the oxidation of methionine units in a ROS environment caused the destabilization of micelles. Such redox- or ROS-triggered micellar destabilization led to enhanced release of encapsulated doxorubicin (DOX) to induce cytotoxicity against cancer cells. Further, the therapeutic effects of the DOX-loaded micelles were demonstrated using the KB cell line. This study shows that thiol and ROS dual-responsive gemini micelles are promising platforms for nano-drug delivery in various cancer therapies.

## 1. Introduction

Biocompatible nanoparticles are promising drug delivery vehicles for anticancer therapy [1]. Particularly, polymeric micelles composed of a hydrophilic shell and a hydrophobic core are popular owing to suitable micelle size, low critical micelle concentration (CMC), and slow rate of dissociation [2,3]. Compared with other nanoparticles, polymeric micelles have been widely studied for cancer chemotherapy because of their remarkable properties, such as the high solubility of hydrophobic drugs in the hydrophobic core, passive targeting ability to tumor tissues through an enhanced permeability and retention (EPR) effect, and prolonged circulation time [4,5,6]. To achieve effective drug release at specific sites as well as reduce damage to normal cells, stimuli-sensitive polymeric micelles have been proposed and fabricated in response to appropriate stimuli such as pH, temperature, light, and a reducing environment [7,8,9]. 

The reductive glutathione (GSH) concentration level is 10 mM in the cytoplasm and around some tumors, while the concentration in the extracellular fluid is as low as 2–20 μM [10,11]. The intracellular GSH can undergo a thiol–disulfide exchange reaction with disulfide bonds and the variation of GSH concentration can provide an opportunity for the design of novel intracellular nanocarriers. Thus, polymer-based nanocarriers containing GSH-cleavable disulfide bonds have been studied as intracellular drug delivery systems [12,13,14,15]. Reactive oxygen species (ROS), including superoxide, hydroxide radical, and hydrogen peroxide, widely exist in living organisms, and play a crucial role in physiological functions, such as regulation of cell signaling pathways, modulation of protein functions, and mediation of inflammation [16,17,18]. It was reported that the ROS hydrogen peroxide is generated during normal metabolism and is produced in large amounts by phagocytic cells at inflammatory sites [19]. Particularly, the concentration of hydrogen peroxide reaches levels around 0.1 mM during inflammation, approximately 100 times higher than that in normal cells [20]. Levels of ROS in tumor cells have also been proposed to have a higher pro-oxidant status [21]. The overproduced-ROS in some diseases and tissues have inspired researchers to develop target-specific drug delivery systems based on ROS. By adopting ROS-responsive linkers, including thioether, selenide, telluride, thioketal, arylboronic ester, oligoproline, and aminoacrylate, various ROS-responsive drug delivery systems have been developed and investigated for therapeutic purposes [22,23,24,25,26,27]. According to the design of the attached linkers, the mechanism of drug release can be ascribed to the change in solubility and cleavage of the linker induced by ROS. For example, Napoli et al. reported a copolymer consisting of poly(ethylene glycol) (PEG) and poly(propylene sulfide) (PPS) for ROS-responsive polymeric vesicles and demonstrated that drug release was caused by the change in solubility due to the phase transition from hydrophobic sulfide to hydrophilic sulfoxide or sulfone in the presence of H_2_O_2_ [28]. In another study, Xia et al. inserted thioketal linker into nanoparticles for ROS sensitivity, as thioketal can be rapidly cleaved by ROS species and degraded into acetone and thiols as by-products [29]. 

Here, we report dual-responsive micelles of a novel gemini amphiphile composed of a thiol-responsive cysteine disulfide spacer and ROS-responsive methionine tails. Gemini or dimeric amphiphile typically show low CMC, better wetting properties, unusual viscosity behavior, and higher solubility compared with the corresponding single chain amphiphiles [30,31,32,33,34]. It is well-known that methionine as a representative antioxidant in vivo acts as a substrate to protect other essential residues, such as cysteine, from ROS damage and can be readily oxidized to methionine sulfoxides by ROS [35,36]. The gemini polymer was characterized by ^1^H NMR, and their micellar properties were investigated using pyrene fluorescence spectroscopy and dynamic light scattering (DLS). The destabilization of gemini micelles in the presence of DTT or H_2_O_2_ was investigated by gel permeation chromatography (GPC) and DLS. The release behavior from doxorubicin (DOX)-loaded micelles was evaluated under H_2_O_2_ or reducing agent. Furthermore, destabilization of DOX-loaded micelles in cellular environments was investigated using confocal laser-scanning microscopy (CLSM) to monitor cellular uptake and cell viability assays were conducted to evaluate anticancer efficacy. 

## 2. Experimental 

### 2.1. Materials 

Methoxypoly(ethylene glycol)-amine (mPEG-NH_2_) with a molecular weight of 2000 g/mol was purchased from JenKem Technology (Beijing, China). *L*-methionine, trichloromethyl chloroformate (TCF), pinene, *N*-(9-fluorenylmethoxycarbonyl)-*S*-trityl-*L*-cysteine (Fmoc-Cys(Trt)-OH), piperidine, *N*,*N*’-dicyclohexylcarbodiimide (DCC), 1,4-dithiothreitol (DTT), trifluoroacetic acid (TFA), doxorubicin hydrochloride (DOX·HCl), glutathione ethyl ester (GSH-OEt), anhydrous tetrahydrofuran (THF), anhydrous dimethylformamide (DMF), and anhydrous dichloromethane (MC) were purchased from Sigma Aldrich (Seoul, Korea) and used as received. KB cells, derived from an epidermal carcinoma of the mouth, were purchased from American Tissue Culture Collection (Seoul, Korea). Eagle’s minimum essential medium (EMEM) was obtained from Life Technologies (Seoul, Korea). Trypsin-EDTA (0.25%) and fetal bovine serum (FBS) were purchased from HyClone Laboratory. The Cell Counting Kit-8 (CCK-8) was obtained from Enzo Life Science (Seoul, Korea).

### 2.2. Measurements

Nuclear magnetic resonance (NMR) spectra were obtained using a Bruker NMR spectrometer (AVANCE III 400) in deuterated chloroform. The molecular weight and polydispersity index of the polymer were determined using gel permeation chromatography (GPC) (1515, Waters) equipped with a Waters 2414 refractive index detector. THF for Waters Styragel HR columns and distilled water containing 0.05 M NaNO_3_ for ultrahydrogel columns were used as the mobile phase with flow rate of 1 mL/min at 35 °C, respectively. The size and distribution of polymeric nanoparticles were measured using dynamic light scattering (DLS, Zetasizer Nano ZS, Malvern Instruments, United Kingdom). Fluorescence spectra were obtained using a Cary Eclipse spectrometer (Varian). 

### 2.3. Preparation of Gemini mPEG-PMT

#### 2.3.1. Methionine N-carboxyanhydride (NCA)

Methionine *N*-carboxyanhydride (NCA): methionine (5 g, 38.11 mmol) was suspended in THF (100 mL). TCF (4.57 mL, 22.87 mmol) and pinene (7.20 mL, 45.73 mmol) were added to the suspension. The reaction temperature was increased to 65 °C, and the reaction was continued until methionine was completely dissolved. Excess hexane was added to the reaction solution to form an oily precipitate, washed with hexane five times, and dried under vacuum. Methionine NCA (3.53 g, 20.15 mmol) was obtained. ^1^H NMR (400 MHz, CDCl3): *δ* (ppm) = 6.87 (broad, 1H, –NH–), 4.52 (d, 1H, –NH*CH*C(O)–), 2.71 (m, 2H, –CH_2_*CH_2_*SCH_3_), 2.31 (m, 2H, –*CH_2_*CH_2_SCH_3_), 2.14 (s, 3H, –CH*_2_*CH_2_S*CH_3_*).

#### 2.3.2. mPEG-Cys(Trt)-Fmoc and mPEG-Cys(Trt)-NH_2_

Both mPEG-Cys(Trt)-Fmoc and mPEG-Cys(Trt)-NH_2_ were prepared as previously described [33]. Briefly, mPEG-Cys(Trt)-Fmoc was prepared by the reaction of mPEG-NH_2_ (2.0 g, 1 mmol) and Fmoc-Cys(Trt)-OH (0.70 g, 1.20 mmol) in anhydrous dichloromethane (60 mL) using DCC (0.25 g, 1.20 mmol) as the coupling agent. The mPEG-Cys(Trt)-NH_2_ (2.0 g) was synthesized by treatment of the mPEG-Cy(Trt)-Fmoc with piperidine in DMF.

#### 2.3.3. mPEG-Cys(Trt)-PMT

mPEG-Cys(Trt)-NH_2_ (1.50 g, 0.65 mmol) as a macroinitiator and methionine NCA (1.35 g, 7.71 mmol) were dissolved in DMF (20 mL) under nitrogen. The polymerization was allowed to proceed for 48 h at room temperature. The solution was precipitated with excess cold diethyl ether, washed with diethyl ether two times, and dried under vacuum. The product (2.24 g) was obtained as a yellow powder with the following characteristics. ^1^H NMR (400 MHz, CDCl_3_): *δ* (ppm) = 7.43, 7.31, and 7.24 ppm (trityl), 3.62~3.57 and 3.40 ppm (mPEG), 3.17 ppm (–HN*CH*C(O)–), 2.97 and 2.91 ppm (–CH*CH_2_*S– of cysteine), 2.69~2.51 ppm (-*CH_2_*SCH_3_ of methionine), 2.23 ppm (–*CH_2_*CH_2_SCH_3_ of methionine), 2.13 ppm (–CH_2_CH_2_S*CH_3_* of methionine).

#### 2.3.4. mPEG-Cys-PMT 

mPEG-Cys(Trt)-PMT (2.0 g) was dissolved in co-solvent consisting of dichloromethane (10 mL) and trifluoroacetic acid (10 mL). Triethylsilane (0.5 mL) was added, and the solution was stirred for 3 h at room temperature. The solvent was evaporated, and the resulting product was precipitated in an excess amount of cold diethyl ether. The precipitate was filtered, washed several times with diethyl ether, and dried under vacuum to obtain 1.81 g of product. ^1^H NMR (400 MHz, CDCl_3_): *δ* (ppm) = 3.62~3.57 and 3.40 ppm (mPEG), 3.17 ppm (–HN*CH*C(O)–), 3.03 and 2.95 ppm (–CH*CH_2_*SH of cysteine), 2.65~2.51 ppm (–*CH_2_*SCH_3_ of methionine), 2.21 ppm (–*CH_2_*CH_2_SCH_3_ of methionine), 2.10 ppm (-CH_2_CH_2_S*CH_3_* of methionine).

#### 2.3.5. Gemini (or dimeric) mPEG-Cys-PMT

For the preparation of the gemini structure, mPEG-Cys-PMT (1.70 g) was dissolved in DMSO (10 mL). The mixture was stirred in an open-air system for 48 h to form disulfide linkages through oxidation between thiols of cysteine. The resulting solution was precipitated in an excess amount of cold diethyl ether. The precipitate was washed several times with diethyl ether and then re-dissolved in dichloromethane (20 mL) to remove monomeric mPEG-Cys-PMT. Trityl chloride resin (0.50 g) was suspended in the solution and the suspension was stirred for 24 h at room temperature. The suspension was filtered out, and the solution was precipitated in excess diethyl ether. The precipitate was dried under vacuum to obtain 0.96 g of gemini mPEG-Cys-PMT. Gemini mPEG-Cys-PMT (0.30 g) was oxidized by treatment with 100 μM H_2_O_2_ for 12 h and 0.31 g of the resulting product was obtained by lyophilization. The chemical structure and molecular weight of gemini mPEG-Cys-PMT treated with or without H_2_O_2_ were analyzed by ^1^H NMR and GPC, respectively.

### 2.4. Micelles Preparation and Characterization

Gemini polymers (10 mg) were dissolved in distilled water (10 mL). The solution was stirred for 6 h at room temperature, yielding micelles with a hydrophobic core in an aqueous solution at 1 g/L. The CMCs of gemini micelles were determined using pyrene as a fluorescence probe. The concentration of polymer varied from 1 × 10^−4^ to 1.0 g/L, and the pyrene concentration was fixed at 0.6 μM. The prepared samples were incubated with stirring at 37 °C for about 36 h to equilibrate the pyrene partition between the water and micelles. Fluorescence spectra were measured using a Varian Cary Eclipse fluorescence spectrometer at an emission wavelength of 390 nm. The CMC was estimated from the inflection point of the intensity ratio I_337_/I_333_ at varied concentrations. The CMCs of monomeric mPEG-Cys-PMT and gemini polymer treated with DTT or H_2_O_2_ were prepared according to the procedures described above and measured by the same method.

### 2.5. ROS- and Thiol-Response Study

The ROS- and thiol-responsive test was performed using gemini mPEG-Cys-PMT alone or mixed with 0.1 mM H_2_O_2_ or 10 mM DTT. After 12 h, the molecular weight and distribution were analyzed by GPC. The stability test was conducted with gemini micelles in phosphate buffered saline (PBS) or mixed with 0.1 mM H_2_O_2_ or 10 mM DTT. After 12 h, the micelle size and distribution were measured by DLS.

### 2.6. Preparation of DOX-Loaded Micelles and Drug Loading Contents 

DOX-loaded micelles were prepared using different feed ratios (Mass_DOX_/Mass_surfactant_ = 0.1, 0.2, 0.3, 0.5, and 1). For preparation of DOX-loaded micelles, DOX (0.5 mg) was dissolved in DMF (0.5 mL) along with polymer (5 mg). Deionized water (5 mL) and TEA (triethylamine, 2 mol equivalent to DOX) was added to the solution while stirring at room temperature. A similar procedure was performed for the predetermined amounts of DOX. After stirring for 6 h, the resulting mixtures were dialyzed (molecular weight cut-off: ~2 kDa) against distilled water for 3 days to remove any unloaded DOX and DMF. The external water was changed twice a day. After dialysis, the solutions were passed through a syringe filter (0.2 μm pore size) to remove large aggregates. DOX-loaded micelles were freeze-dried and re-dissolved in DMF (4 mL), followed by fluorescence spectral analysis. A calibration curve was obtained using various concentrations of DOX in DMF, and the DOX loading content was calculated by the weight ratio of loaded DOX to dried sample, as shown in Appendix A.

### 2.7. Release of DOX from DOX-Loaded Micelles

DOX release from DOX-loaded micelles was investigated at 37 °C in phosphate buffered saline (PBS, 10 mM, pH 7.4). The sample was prepared as mentioned for determination of DOX loading content. Aliquots of DOX-loaded poly(gemini) micellar dispersion (Mass_DOX_/Mass_surfactant_ = 0.5, 10 mL) were introduced into a dialysis tube (MWCO: ~2 kDa). The dialysis bags were immersed in PBS buffer solution (100 mL) as a control and aqueous 10 mM DTT or 0.1 mM H_2_O_2_ solution buffered with PBS. The release medium was shaken at 120 rpm at 37 °C. At pre-determined intervals, 3 mL samples were withdrawn from the release medium, and an equivalent volume of fresh medium was subsequently added. The DOX concentrations were determined by a fluorescence spectrometer at an excitation wavelength of 485 nm and an emission wavelength of 550 nm.

### 2.8. Cytotoxicity

KB cells were grown in EMEM supplemented with 10% FBS. The cells were maintained at 37 °C and 5% CO_2_ in a humidified incubator and the medium was changed every other day. To determine the cytotoxicity of gemini mPEG-Cys-PMT micelles, KB cells were seeded at a density of 1 × 10^3^ cells/well in 96-well plates in 100 μL medium and incubated for 24 h. Next, the cells were treated with micelles at different concentrations (from 0 to 100 μg/mL) in PBS for 48 h. Cells cultured with PBS (HyClone Laboratory) were used as a control. To measure the cytotoxicity, CCK-8 was used following the manufacturer’s instructions. Briefly, 10 μL of CCK-8 reagent was added into each well and the plate was incubated at 37 °C for 2 h. The absorbance was detected at 450 nm using a Multiskan microplate reader (Thermo Fisher Scientific Inc., Waltham, MA, USA). The cytotoxicity was expressed as percentage of viable cells relative to the viability of the control cells. All of the experiments were performed in triplicate, and the data are represented as means ± S.D. To estimate the effect of GSH or H_2_O_2_ on the cytotoxicity of DOX-loaded micelles, the intracellular level of thiol or ROS was manipulated by adding GSH-OEt and H_2_O_2_ into the cell culture media, respectively. KB cells (100 μL) were evenly seeded into a 96-well plate at a density of 1 × 10^3^ cells/well and incubated for 24 h. The cells were first treated with 10 mM GSH-OEt or 0.1 mM H_2_O_2_ for 3 h, followed by washing with PBS and adding fresh culture media. DOX-loaded micelles were added to meet the DOX concentrations of 0, 0.4, 0.8, 1.2, and 1.6 μM for 0, 1, and 10 mM GSH-OEt or 0.1 mM H_2_O_2_ pre-treated cells, respectively. After incubation for 24 h, the cytotoxicity of the DOX-loaded micelles was evaluated using the CCK-8 assay. For reference, free DOX was also tested for cytotoxicity.

### 2.9. Intracellular DOX Release

To observe the cellular uptake of DOX-loaded micelles, KB cells were seeded in a 35 mm glass-bottom dish (SPL Life Science, Pocheon-Si, Korea) at a density of 5 × 10^4^ cells/mL and cultured for 24 h. The cells were first treated with 10 mM GSH-OEt or 0.1 mM H_2_O_2_ for 3 h. The medium was replaced, and DOX-loaded micelles were added to the cells at a concentration of 2.5 μM. After 3 h, the cells were washed three times with PBS and stained with 4’,6-diamidino-2-phenylindole (DAPI). Then, cells were fixed by exposure to 2.5% glutaraldehyde in PBS for 1 h before observation. The cells were observed under a confocal laser-scanning microscope (FV1200, Olympus, Nagano, Japan) by excitation at 405 nm under the same laser light intensity and gain value between samples. The emission was measured at 461 and 564 nm for DAPI and DOX, respectively.

## 3. Results and Discussion

### 3.1. Synthesis and Micelles

The dual-responsive gemini amphiphile composed of hydrophilic methoxy poly(ethylene glycol) (mPEG) and hydrophobic poly(methionine) (PMT) containing cysteine linkage as connecting group was prepared, as shown in Scheme 1. The mPEG-Cys(Trt)-NH_2_ was first synthesized by the condensation of mPEG-NH_2_ and Fmoc-Cys(Trt)-OH, following the deprotection of the Fmoc group with piperidine. The mPEG-Cys-PMT was prepared by ring-opening polymerization of methionine-NCA with mPEG-Cys(Trt)-NH_2_ as macroinitiator and subsequent deprotection of the trityl group. Gemini (or dimeric) mPEG-Cys-PMT were synthesized by oxidation between thiols of cysteine in DMSO, and then purified by treatment with trityl chloride resin to remove monomeric mPEG-Cys-PMT. The synthesized gemini mPEG-Cys-PMT were characterized by ^1^H NMR, as shown in Figure 1A. The strong peaks at 3.40 and 3.48~3.74 ppm were assigned to the methyl group (CH_3_O–) and PEG (–OCH_2_CH_2_–), respectively. The weak peaks at 3.03 and 2.94 ppm were attributed to the methylene groups close to the disulfide linkage (–CH_2_SSCH_2_–). The chemical shift at 2.51~2.65 ppm and 2.14 ppm assigned to the methylene groups of methionine (–CH_2_CH_2_SCH_3_), and the strong resonance peak at 2.10 ppm to the methyl group (–CH_2_CH_2_SCH_3_) of methionine demonstrated the successful ring opening polymerization of methionine NCA. The degree of polymerization (DP) for gemini mPEG-Cys-PMT was characterized by ^1^H NMR and calculated from the integral ratio of the methyl group of mPEG at 3.40 ppm (a, CH_3_O–) to that of methionine at 2.10 ppm (g, CH_3_SCH_2_CH_2_–); a DP value of 9.84 was obtained. The oxidized gemini mPEG-Cys-PMT obtained by treatment with 0.1 mM H_2_O_2_ for 12 h was evaluated by ^1^H NMR analysis through changes in the chemical shift from thioether to sulfoxide or sulfone, as shown in Figure 1B. After oxidation with 0.1 mM H_2_O_2_, the methylene peak (–CH_2_CH_2_SCH_3_) close to the thioether at 2.14 ppm almost disappeared over 24 h whereas a proton peak close to the sulfoxide peak at 2.74 ppm emerged. Peak analysis indicates that the thioether groups of gemini mPEG-Cys-PMT reacted with ROS and were oxidized to form sulfoxide or sulfone groups. GPC measurements of monomeric and gemini (dimeric) mPEG-Cys-PMT were performed using THF as a mobile phase. The number-average molecular weight increased from 3750 g/mol to 6870 g/mol due to oxidation between cysteine residues of monomeric mPEG-Cys-PMT, confirming the formation of the gemini structure, as shown in Appendix A. The molecular weights calculated by ^1^H NMR and determined by GPC were comparable.

The CMC value of gemini mPEG-Cys-PMT, measured by fluorescence spectroscopy using pyrene as a fluorescent probe was 7.24 × 10^−3^ mg/L, as shown in Figure 2A. Here, H_2_O_2_ was used to make the ROS environment for investigating the ROS-dependent behavior of gemini mPEG-Cys-PMT. In the presence of H_2_O_2_, we hypothesized that the hydrophobic thioether groups of PMT could be converted into hydrophilic sulfoxide or sulfone that are soluble or swell in water. After oxidation with 0.1 mM H_2_O_2_ for gemini mPEG-Cys-PMT, the CMC of oxidized gemini mPEG-Cys-PMT was determined to be 8.13 × 10^−2^ mg/L, as shown in Figure 2B. This result shows that the CMC value was increased owing to the increase in hydrophilicity caused by the formation of hydrophilic sulfoxide or sulfone. Furthermore, it is expected that the release of an encapsulated drug can be easily achieved through change in CMC upon the formation of hydrophilic sulfoxide or sulfone in the presence of ROS. The CMCs of monomeric mPEG-Cys-PMT and gemini mPEG-Cys-PMT treated DTT were 2.40 × 10^−2^ and 1.80 × 10^−2^ mg/L, respectively, as shown in Appendix A. The CMC of the gemini micelles was much lower than that of the monomeric micelles. Also, the CMC of gemini micelles treated with DTT was higher than that of gemini micelles, which can be attributed to the formation of monomeric mPEG-Cys-PMT after the cleavage of cysteine spacer under DTT.

### 3.2. ROS- and Redox-Responsiveness

For gemini mPEG-Cys-PMT treated with H_2_O_2_, ROS-responsiveness was evaluated by examining the change in molecular weight distribution by GPC measurement, as shown in Figure 3A. GPC measurement was performed using distilled water containing 0.05 M NaNO_3_ as a mobile phase because of non-solubility of the oxidized gemini mPEG-Cys-PMT in THF after the treatment of H_2_O_2_. The molecular weight increased owing to the oxidation of methionine units of gemini mPEG-Cys-PMT, confirming the successful formation of sulfoxide. Thiol-responsiveness for cysteine disulfide linkage as spacer was measured by GPC after treatment with 10 mM DTT for gemini mPEG-Cys-PMT and oxidized gemini form. The decrease in molecular weight in the GPC traces indicated that gemini (dimeric) structure was converted into the corresponding monomeric structure and its DTT adducts through a thiol–disulfide exchange reaction with DTT. From these results, it was confirmed that gemini mPEG-Cys-PMT composed of cysteine linkage as spacer and methionine units as tail have dual-sensitivity under reducing or ROS environments. Further, size and distribution of gemini mPEG-Cys-PMT micelles under redox or ROS environment were examined. The cytosolic GSH level in some tumor cells was increased at least four times than that in normal cells and the concentration of H_2_O_2_ in the tumor cellular milieu was increased to 0.1 mM. The responsiveness of gemini mPEG-Cys-PMT micelles to these conditions was evaluated by setting concentrations similar to the tumor cellular environment. Figure 3B shows particle size and distribution of gemini mPEG-Cys-PMT micelle in presence of a redox or ROS environment. The particle size and PDI (polydispersity index) of gemini mPEG-Cys-PMT micelle were measured as 178.6 nm and 0.13, respectively. In 10 mM DTT as the redox surrounding, the population of origin aggregates decreased to 153.7 nm and the distribution became broader. This result indicated that gemini mPEG-Cys-PMT micelles were destabilized upon cleavage of cysteine spacer in response to a redox environment and were rearranged into monomeric micelles. When 0.1 mM H_2_O_2_ was added to gemini mPEG-Cys-PMT micelles, the size distribution became bimodal, with the occurrence of small aggregates of approximately 60 nm size. This is attributed to the formation of hydrophilic sulfoxide as a result of the oxidation of the thioether with the addition of H_2_O_2_. In the presence of co-agents such as 10 mM DTT and 0.1 mM H_2_O_2_, similar results were obtained to those when only DTT was treated, showing the slight decrease in mean size and broadening of the distribution. This result may be due to higher DTT concentration than H_2_O_2_. 

### 3.3. Drug Loading and Release

The drug loading contents and loading efficiency of gemini mPEG-Cys-PMT micelles were determined using DOX as a hydrophobic drug with polymer:DOX feed weight ratios varying from 1:0.1 to 1:1 at a polymer concentration of 0.1 mg/mL, as shown in Appendix A. When the feed ratio of polymer to DOX was 1:0.5, the maximum loading content in gemini mPEG-Cys-PMT micelles and corresponding drug loading efficiency were 6.35% and 19.07%, respectively. The gemini mPEG-Cys-PMT micelles showed slightly increased DOX loading content while their loading efficiency decreased with the increase in DOX feed weight ratio. The release behaviors of DOX-loaded micelles in DTT or H_2_O_2_ were investigated to clarify whether micelles can release loaded-DOX. The DOX-loaded micelles were treated with 10 mM DTT, 0.1 mM H_2_O_2_ or DTT and H_2_O_2_, and DOX release was monitored at a pre-determined time interval for 24 h, as shown in Figure 4. In the absence of DTT and H_2_O_2_ as a control, less than 21% of the encapsulated DOX was released over 24 h, suggesting a release by diffusion from intact micellar aggregates rather than from disassembly of micelles by stimulus because DOX was confined in the hydrophobic core of micelles. In the presence of DTT, the DOX release from DOX-loaded micelles increased to approximately 78% in 10 mM DTT over 24 h. In the DTT condition, gemini micelles containing cysteine disulfide attached as a spacer can undergo a thiol–disulfide exchange reaction, and DOX release is accelerated due to destabilization of micelles. Moreover, in the presence of H_2_O_2_, the release of DOX was obviously accelerated to 69% under 0.1 mM H_2_O_2_ condition over 24 h, proving that methionine units containing ROS-responsive thioether groups can react well with H_2_O_2_. As shown in the previous CMC result, the oxidation of methionine chains under ROS can lead to the solubility conversion and destabilization of micelle, and DOX is then efficiently released. Comparing the release behavior in a redox environment with that of ROS for the initial 4 h, the DOX was released faster in the presence of DTT and the release rates for DOX-loaded micelles in each environment increased with time. Under the mixture of 10 mM DTT and 0.1 mM H_2_O_2_, the DOX release behavior was almost similar to that when only DTT was used. It is well known that thiol molecules in living organisms play a role as a barrier against ROS [35]. These comparable results were expected because a relatively high DTT concentration could act as a protective barrier against ROS. Thus, the DOX release from DOX-loaded micelles in a redox surrounding was more dominant than that in a ROS surrounding. From the release results of the encapsulated DOX in a tumor-like environment, gemini mPEG-Cys-PMT micelles were confirmed to have dual responsiveness.

### 3.4. Cytotoxicity of Micelles

Cell viability of the gemini mPEG-Cys-PMT micelles was investigated using the CCK-8 assay. KB cells were cultured with different concentrations of micelles and then incubated for 24 h. Figure 5A suggests high viability (>95%) of cells, with no significant difference compared with control cells, indicating that gemini mPEG-Cys-PMT micelles were not cytotoxic to cells at concentrations up to 0.1 mg/mL, and thus are biocompatible. The cytoprotective effect of the gemini mPEG-Cys-PMT micelles was evaluated against cytotoxic levels of H_2_O_2_. KB cells were seeded on culture plate and incubated for 24 h. KB cells were treated with varying concentrations of H_2_O_2_ (0.1, 0.5, and 1.0 mM H_2_O_2_) for 3 h, and then incubated for additional 24 h. Additionally, cells were treated with gemini mPEG-Cys-PMT micelles and incubated under different conditions by addition of H_2_O_2_ for 24 h. As shown in Figure 5B, in absence of micelles, the cells incubated with 0.1 mM H_2_O_2_ showed high viability. Notably, the viability of cells treated with 0.5 and 1 mM H_2_O_2_ decreased to 48% and 6.3%, respectively, indicating a high cytotoxicity in the H_2_O_2_-rich environment. The viability in the presence of micelles, compared to that in presence of H_2_O_2_ alone, increased by 87% and 61% when the cells were treated with 0.5 mM and 1 mM H_2_O_2_, respectively. These results showed that the cytotoxicity of H_2_O_2_ could be reduced because of the scavenging effect derived from the oxidation of thioethers in methionine units by H_2_O_2_. Besides, the cytoprotective properties of the gemini mPEG-Cys-PMT supported their potential application in therapies for diseases caused by a higher level of ROS. The treatment with 10 mM GSH-OEt alone or the mixture (10 mM GSH-OEt and 0.1 mM H_2_O_2_) for 24 h did not affect the viability of KB cells. Moreover, after cells were treated with micelles, the addition of 10 mM GSH-OEt alone or the mixture did not affect the viability of cells.

### 3.5. Cytotoxicity of Drug-Loaded Micelles and Internalization

The cytotoxicity of DOX-loaded gemini micelles upon treatment with GSH or H_2_O_2_ was examined against the KB cell line. KB cells were pre-treated with and without 10 mM GSH-OEt or 0.1 mM H_2_O_2_ for 3 h and then incubated with various concentrations of DOX-loaded gemini micelles for 24 h. GSH-OEt, a neutralized form of GSH, is known to penetrate cellular membranes and hydrolyze in the cytoplasm to generate GSH [34]. Several studies reported the pre-treatment of cancer cells with GSH-OEt to enhance GSH levels [37,38,39]. Similarly, we pre-treated cancer cells with 0.1 mM H_2_O_2_ to enhance ROS concentration. As a control for comparison, cells were also treated with DOX. As seen in Figure 6A, the viability of KB cells decreased with increasing concentration of DOX and encapsulated-DOX, showing inhibition of cell proliferation in presence of DOX. Compared with no GSH-OEt and H_2_O_2_ pre-treatment conditions, the viability was higher when KB cells were treated with GSH-OEt and H_2_O_2_. This result indicated that the intracellular GSH concentration increased in KB cells by GSH-OEt pre-treatment and facilitated the cytosolic release of DOX from DOX-loaded poly(gemini) micelles, resulting in structural change by degradation of poly(gemini) micelles into monomeric structure through the cleavage of the intra-molecular disulfide bond. In our experiment, 0.1 mM H_2_O_2_ was not cytotoxic to KB cells. The significant reduction in cells pre-treated with H_2_O_2_ was attributed to the presence of more H_2_O_2_ in KB cells, which causes the enhanced release of DOX upon oxidation of thioether to inhibit the proliferation of KB cells. The DOX-loaded micelles without GSH-OEt or H_2_O_2_ treatment showed cytotoxicity, which could be due to the presence of GSH and H_2_O_2_ in KB cells triggering the destabilization of DOX-loaded micelles. The cellular uptake and intracellular release of DOX in response to cellular GSH or ROS for KB cells were studied using CLSM. KB cells were pre-treated with 10 mM GSH-OEt and 0.1 mM H_2_O_2_ for 3 h and then incubated with DOX-loaded micelles and free DOX for 3 h. Localization of DOX within the cells was evaluated using the red auto-fluorescence from DOX and the blue fluorescence from DAPI. Figure 6B shows the distribution of DOX in cells. Free DOX was observed in the nucleus, indicating that it could penetrate the nuclear membranes. The cells incubated with only DOX-loaded micelles emitted weak red DOX fluorescence in the nucleus. However, the intensity of DOX in the nucleus of KB cells pre-treated with GSH-OEt or H_2_O_2_ was brighter than that of untreated cells. These results indicate that the encapsulation of DOX into micelles caused retardation of DOX release, but the degradation mediated by GSH-OEt or H_2_O_2_ accelerated the intracellular DOX release from DOX-loaded micelles. The results from cell viability measurements and CLSM together suggest faster DOX release from destabilization of DOX-loaded micelles by higher intracellular GSH or H_2_O_2_ concentration, enhancing the inhibition of cellular proliferation after internalization into cells.

## 4. Conclusions

A dual-responsive gemini amphiphile consisting of hydrophilic poly(ethylene glycol) (PEG) blocks and hydrophobic methionine units joined by cystine disulfide spacer was successfully synthesized and self-assembled to form colloidal stable micellar aggregates in aqueous solutions. Their dual-responsive properties were verified by DLS and GPC analysis. Cystine disulfide linkage as spacer was cleaved under a reductive environment and methionine units as tails were oxidized in the presence of ROS, resulting in degradation or destabilization of micelles. Cell viability and CLSM studies demonstrated that intracellular release of DOX from DOX-loaded micelles after internalization into KB cells enhanced in response to a high level of ROS or GSH. These results suggest that GSH- or ROS-responsive micelles hold great promise as anticancer drug carrier platforms for reducing side effects and improving the efficacy of anticancer treatment.

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
