# Peer review of "Dual-responsive Gemini Micelles for Efficient Delivery of Anticancer Therapeutics"

_polymers, 2019, doi:10.3390/polym11040604_

Round 1

Reviewer 1 Report

I greatly enjoyed reading the manuscript, the content was delivered in clear and concise manner. I have only minor comment for the improvement of the manuscript.

1.      Line 89     Please describe KB cells in detail. 

2.      Figure 4    The third curve labeling misses “H2O2”.

3.      Figure 6B   Need to add bar

Author Response

1.   Line 89     Please describe KB cells in detail. 

  On line 89, a further description of the KB cell was inserted.

2.   Figure 4    The third curve labeling misses “H2O2”.

   The labeling error in the Figure 4 has been corrected and added.

3.    Figure 6B   Need to add bar

  We inserted the scale bar in the Figure 6 (B).

Reviewer 2 Report

The authors have prepared novel polymers for ROS and GSH responsiveness. However, here are some suggestions: 1. The polymeric micelles spontaneously formed in aqueous medium. However, in this manuscript, there is no evidence to prove the micellar structure. The TEM or SEM images were essential. 2. For the responsive tests, only polymers were characterized. The micelles composed of the polymers should also be tested for the responsive tests. The particle sizes of the polymeric micelles might be different in high levels of ROS or GSH conditions. 3. In this study, the KB cells were treated with H2O2 or GSH for the cell viability or intracellular drug releasing tests, while the cancer cells would generate GSH in the cytoplasm within cells. Besides, the high levels of ROS within cancer cells were also reported. Herein, the cellular experiments should not add H2O2 or GSH in cell medium. The extra H2O2 or GSH might interfere the cell experiments. 4. The structure of this article must be improved. For example, the introduction should not mentioned the results. The section for materals and methods should not include 1H-NMR peak (section 2.3.1, 2.3.3 and 2.3.4) and the characterization should clearly interpret in every procedure (section 2.3.2). 5. In Figure 4, 100 uM H2O2 should be present. 6. In section 2.1, the CLSM machine should also be involved. 7. Line 109-110, the sentence " The solution was precipitated with excess hexane, washed with hexane 110 five times, and dried under vacuum, after which a brownish oil, methionine NCA" was confusing.

Author Response

Thank for your comments.

1. The polymeric micelles spontaneously formed in aqueous medium. However, in this manuscript, there is no evidence to prove the micellar structure. The TEM or SEM images were essential.

The data on critical micelle concentration, one of the main features of polymeric micelles, are shown in Figure 2 and supplementary information (Figure S3).

2. For the responsive tests, only polymers were characterized. The micelles composed of the polymers should also be tested for the responsive tests. The particle sizes of the polymeric micelles might be different in high levels of ROS or GSH conditions.

The responsive tests of polymeric micelles were performed under ROS or DTT and is shown in Figure 3 (B).

3. In this study, the KB cells were treated with H2O2 or GSH for the cell viability or intracellular drug releasing tests, while the cancer cells would generate GSH in the cytoplasm within cells. Besides, the high levels of ROS within cancer cells were also reported. Herein, the cellular experiments should not add H2O2 or GSH in cell medium. The extra H2O2 or GSH might interfere the cell experiments.

The DOX-micelles labeling in Figure 6 (A) shows the experimental data without adding H2O2 or GSH.

4. The structure of this article must be improved. For example, the introduction should not mentioned the results. The section for materals and methods should not include 1H-NMR peak (section 2.3.1, 2.3.3 and 2.3.4) and the characterization should clearly interpret in every procedure (section 2.3.2).

We removed the results mentioned in the introduction according to your advice (line 69-76).

Before correction: Monomeric mPEG-Cys-PMT was prepared by polymerization of methionine carboxyanhydride (NCA) and deprotection of trityl group. Gemini mPEG-Cys-PMT was synthesized by oxidation between cysteine of corresponding monomeric amphiphile. The gemini mPEG-Cys-PMT was characterized by 1H NMR, and their micellar properties were investigated using pyrene fluorescence spectroscopy and dynamic light scattering (DLS). The destabilization of gemini micelles in the presence of DTT or H2O2 was investigated by gel permeation chromatography (GPC) and DLS; such micellar destabilization led to enhanced release of encapsulated DOX, which was used as model anticancer drug. Furthermore, destabilization of DOX-loaded micelles in cellular environments was investigated using confocal laser-scanning microscopy (CLSM) to monitor cellular uptake and cell viability assays were conducted to evaluate anticancer efficacy. The obtained results indicated that gemini mPEG-Cys-PMT micelles had several advantages and was a promising drug carrier for cancer treatment.

After correction: The gemini polymer was characterized by 1H NMR, and their micellar properties were investigated using pyrene fluorescence spectroscopy and dynamic light scattering (DLS). The destabilization of gemini micelles in the presence of DTT or H2O2 was investigated by gel permeation chromatography (GPC) and DLS. The release behavior from DOX-loaded micelles was evaluated under H2O2 or reducing agnet.  Furthermore, destabilization of DOX-loaded micelles in cellular environments was investigated using confocal laser-scanning microscopy (CLSM) to monitor cellular uptake and cell viability assays were conducted to evaluate anticancer efficacy.

 According to your comment, I do not think the subtitle (materials and methods) of this section is appropriate, so I revised this section as a subtitle of experimental.

5. In Figure 4, 100 uM H2O2 should be present.

The labeling error in the Figure 4 has been corrected and added.

 6. In section 2.1, the CLSM machine should also be involved.

In section 2.9, we have already described the specifications and experimental condition of CLSM machine.

7. Line 109-110, the sentence " The solution was precipitated with excess hexane, washed with hexane 110 five times, and dried under vacuum, after which a brownish oil, methionine NCA" was confusing.

The confused part was modified to the following expression.Excess hexane is added to the reaction solution to form an oily precipitate, washed with hexane five times, and dried under vacuum.

Reviewer 3 Report

In this manuscript, the authors developed reduction and reactive oxygen species dual responsive micelles for anti-cancer delivery. The intracellular environment sensitive drug release behavior is beneficial to reduce the premature drug release during circulation, and thus reduce the side effects of drugs to normal tissues. Overall, the material is of good novelty with the dual sensitivity to DTT and ROS. And the characterization of the materials and micelles is sufficient to demonstrate the successful synthesis of the material and good properties of the micelles for drug delivery. In the in vitro anti-cancer treatment with KB cells, the drug delivery system also exhibited good anti-cancer efficacy.

One major question I have for this manuscript is the material design. As can be seen in Figure 4, the drug release of this system in the presence of both DTT and hydrogen peroxide doesn’t show great improvement compared with the release behavior after addition of DTT or hydrogen peroxide alone.  And at some time points, the release can be even slower than that of DTT addition alone. That might be an indicator showing that DTT and ROS work against each other, DTT works through reduction of the disulfide bonds while the ROS works through oxidation. So it would be good if the authors could make some comments here.

Minor problems:

Figure 4, "100uM" should be changed to "100 uM H2O2", and more time points in the release experiment could be obtained if possible (so we can see the plateau of the release curves) 

In Figure 6B, a scale bar should be provided. The DOX fluorescence intensity between the different groups are not significant enough to support the conclusions. Some bright DOX fluorescence dots could be observed in Micelles/DOX group. 

But overall, this delivery system is responsive to either DTT or ROS for drug release and exhibit good anti-cancer property.       

Author Response

One major question I have for this manuscript is the material design. As can be seen in Figure 4, the drug release of this system in the presence of both DTT and hydrogen peroxide doesn’t show great improvement compared with the release behavior after addition of DTT or hydrogen peroxide alone.  And at some time points, the release can be even slower than that of DTT addition alone. That might be an indicator showing that DTT and ROS work against each other, DTT works through reduction of the disulfide bonds while the ROS works through oxidation. So it would be good if the authors could make some comments here.

Thank for your advice. In this regard, it is thought that more research should be done in the future. In brief, it is thought that the concentration of DTT is relatively higher that that of hydrogen peroxide, so it is similar to that of DTT alone.

Minor problems:

Figure 4, "100uM" should be changed to "100 uM H2O2", and more time points in the release experiment could be obtained if possible (so we can see the plateau of the release curves) 

The labeling error in the Figure 4 has been corrected and added.

In Figure 6B, a scale bar should be provided. The DOX fluorescence intensity between the different groups are not significant enough to support the conclusions. Some bright DOX fluorescence dots could be observed in Micelles/DOX group. 

We inserted the scale bar in the Figure 6 (B).

Round 2

Reviewer 2 Report

The authors have reviewed and answered my questions. The dual ROS and GSH responsive micelles possessed potentials in anti cancer therapy.

Author Response

1. The polymeric micelles spontaneously formed in aqueous medium. However, in this manuscript, there is no evidence to prove the micellar structure. The TEM or SEM images were essential.

The data on critical micelle concentration, one of the main features of polymeric micelles, are shown in Figure 2 and supplementary information (Figure S3).

2. For the responsive tests, only polymers were characterized. The micelles composed of the polymers should also be tested for the responsive tests. The particle sizes of the polymeric micelles might be different in high levels of ROS or GSH conditions.

The responsive tests of polymeric micelles were performed under ROS or DTT and is shown in Figure 3 (B).

3. In this study, the KB cells were treated with H2O2 or GSH for the cell viability or intracellular drug releasing tests, while the cancer cells would generate GSH in the cytoplasm within cells. Besides, the high levels of ROS within cancer cells were also reported. Herein, the cellular experiments should not add H2O2 or GSH in cell medium. The extra H2O2 or GSH might interfere the cell experiments.

The DOX-micelles labeling in Figure 6 (A) shows the experimental data without adding H2O2 or GSH.

4. The structure of this article must be improved. For example, the introduction should not mentioned the results. The section for materals and methods should not include 1H-NMR peak (section 2.3.1, 2.3.3 and 2.3.4) and the characterization should clearly interpret in every procedure (section 2.3.2).

We removed the results mentioned in the introduction according to your advice (line 69-76).

Before correction: Monomeric mPEG-Cys-PMT was prepared by polymerization of methionine carboxyanhydride (NCA) and deprotection of trityl group. Gemini mPEG-Cys-PMT was synthesized by oxidation between cysteine of corresponding monomeric amphiphile. The gemini mPEG-Cys-PMT was characterized by 1H NMR, and their micellar properties were investigated using pyrene fluorescence spectroscopy and dynamic light scattering (DLS). The destabilization of gemini micelles in the presence of DTT or H2O2 was investigated by gel permeation chromatography (GPC) and DLS; such micellar destabilization led to enhanced release of encapsulated DOX, which was used as model anticancer drug. Furthermore, destabilization of DOX-loaded micelles in cellular environments was investigated using confocal laser-scanning microscopy (CLSM) to monitor cellular uptake and cell viability assays were conducted to evaluate anticancer efficacy. The obtained results indicated that gemini mPEG-Cys-PMT micelles had several advantages and was a promising drug carrier for cancer treatment.

After correction: The gemini polymer was characterized by 1H NMR, and their micellar properties were investigated using pyrene fluorescence spectroscopy and dynamic light scattering (DLS). The destabilization of gemini micelles in the presence of DTT or H2O2 was investigated by gel permeation chromatography (GPC) and DLS. The release behavior from DOX-loaded micelles was evaluated under H2O2 or reducing agnet.  Furthermore, destabilization of DOX-loaded micelles in cellular environments was investigated using confocal laser-scanning microscopy (CLSM) to monitor cellular uptake and cell viability assays were conducted to evaluate anticancer efficacy.

According to your comment, I do not think the subtitle (materials and methods) of this section is appropriate, so I revised this section as a subtitle of experimental.

5. In Figure 4, 100 uM H2O2 should be present.

The labeling error in the Figure 4 has been corrected and added.

 6. In section 2.1, the CLSM machine should also be involved.

In section 2.9, we have already described the specifications and experimental condition of CLSM machine.

7. Line 109-110, the sentence " The solution was precipitated with excess hexane, washed with hexane 110 five times, and dried under vacuum, after which a brownish oil, methionine NCA" was confusing.

The confused part was modified to the following expression.Excess hexane is added to the reaction solution to form an oily precipitate, washed with hexane five times, and dried under vacuum.